# Regulatory Mechanism of the IL-33–IL-37 Axis via Aryl Hydrocarbon Receptor in Atopic Dermatitis and Psoriasis

**DOI:** 10.3390/ijms241914633

**Published:** 2023-09-27

**Authors:** Gaku Tsuji, Kazuhiko Yamamura, Koji Kawamura, Makiko Kido-Nakahara, Takamichi Ito, Takeshi Nakahara

**Affiliations:** 1Research and Clinical Center for Yusho and Dioxin, Kyushu University Hospital, 3-1-1, Maidashi, Higashi-ku, Fukuoka 812-8582, Japan; yamamura.kazuhiko.821@m.kyushu-u.ac.jp (K.Y.); nakahara.takeshi.930@m.kyushu-u.ac.jp (T.N.); 2Department of Dermatology, Graduate School of Medical Sciences, Kyushu University, 3-1-1, Maidashi, Higashi-ku, Fukuoka 812-8582, Japan; kokawamu@gmail.com (K.K.); nakahara.makiko.107@m.kyushu-u.ac.jp (M.K.-N.); ito.takamichi.657@m.kyushu-u.ac.jp (T.I.)

**Keywords:** IL-33, IL-37, atopic dermatitis, psoriasis

## Abstract

Interleukin (IL)-33 and IL-37 have been identified as novel cytokines involved in various inflammatory diseases. However, their specific roles remain largely unknown. Recent studies have shown that IL-33, which triggers inflammation, and IL-37, which suppresses it, cooperatively regulate the balance between inflammation and anti-inflammation. IL-33 and IL-37 are also deeply involved in the pathogenesis of inflammatory skin diseases such as atopic dermatitis (AD) and psoriasis. Furthermore, a signaling pathway by which aryl hydrocarbon receptor (AHR), a receptor for dioxins, regulates the expression of IL-33 and IL-37 has been revealed. Here, we outline recent findings on the mechanisms regulating IL-33 and IL-37 expression in AD and psoriasis. IL-33 expression is partially dependent on mitogen-activated protein kinase (MAPK) activation, and IL-37 has a role in suppressing MAPK in human keratinocytes. Furthermore, IL-33 downregulates skin barrier function proteins including filaggrin and loricrin, thereby downregulating the expression of IL-37, which colocalizes with these proteins. This leads to an imbalance of the IL-33–IL-37 axis, involving increased IL-33 and decreased IL-37, which may be associated with the pathogenesis of AD and psoriasis. Therefore, AHR-mediated regulation of the IL-33–IL-37 axis may lead to new therapeutic strategies for the treatment of AD and psoriasis.

## 1. Introduction

Interleukin (IL)-33 and IL-37 have been identified as novel cytokines involved in various inflammatory diseases [1,2]. However, their specific roles remain largely unknown. Recent studies have suggested that IL-33 and IL-37 are new potential therapeutic targets in inflammatory skin diseases [3,4]. Furthermore, IL-33 and IL-37 expression are coordinately regulated, and the IL-33–IL-37 axis has been shown to regulate the balance between inflammation and anti-inflammation [5,6]. It has also been shown that the aryl hydrocarbon receptor (AHR), a receptor for dioxins, is involved in the pathogenesis of inflammatory skin diseases such as atopic dermatitis (AD), psoriasis [7,8], hidradenitis suppurativa (HS), and acne [9]. AHR contributes to the pathogenesis of these inflammatory skin diseases by regulating the production of inflammatory cytokines such as TNF-α and IL-8 [9] and the Th17 cytokine network [9]. In addition, it has been reported that the ligands of AHR have two opposing sets of properties, one that aggravates inflammation and one that suppresses inflammation [10].

Against this background, this review outlines the latest findings on the regulatory mechanisms of the IL-33–IL-37 axis by AHR in AD and psoriasis.

## 2. Structure and Signaling Pathway of IL-33

IL-33 is a cytokine belonging to the IL-1 family, previously named IL-1F11 [11]. IL-33 has been reported to be expressed on keratinocytes [12], fibroblasts [12,13], vascular endothelial cells [12], dendritic cells [14], macrophages [15], and mast cells [16]. IL-33 localizes to the nucleus of these cells as an immature form (full-length IL-33) consisting of 270 amino acids and bound to chromatin-associated histones via its N-terminal chromatin-binding domain [17]. When epithelial cells are exposed to allergens, a ripoptosome containing caspase-8 is formed intracellularly, followed by the activation of caspases-3 and -7. This cleaves the immature form to form the active form, which is secreted out of the cells [18]. In addition to this intracellular pathway, allergen-derived extracellular proteases and proteases from neutrophils and mast cells cleave the protease-sensing domain of IL-33 to generate active IL-33 [18,19]. The active form of IL-33 is also secreted extracellularly when cells undergo cell death (necrosis) due to physical damage (e.g., scratching injury) or infection [20].

Two types of IL-33 receptors have been identified: transmembrane suppression of tumorigenicity (ST) 2L receptor and secreted ST2 (soluble ST2) [21]. ST2L forms a dimer with IL-1RAcP, and the binding of IL-33 to it transduces inflammatory signals into the cell, including Th2 cells, regulatory T cells, type 2 innate lymphocytes (ILC2), dendritic cells, macrophages, eosinophils, basophils, and mast cells [22]. In contrast, soluble ST2 lacks an intracellular signaling domain and is thought to act as a decoy receptor to suppress inflammatory signaling by IL-33 [22]. Binding of IL-33 to ST2L receptor induces myeloid differentiation factor 88 (MyD88)-dependent activation of nuclear factor-kappa B (NF-κB) and mitogen-activated protein kinase (MAPK), which regulate gene expression [23].

The mechanism by which cytokines regulate IL-33 expression has remained largely unknown, but it depends on the type of cell on which the cytokines act [24]. It has been reported that tumor necrosis factor (TNF)-α induces the production of IL-33 in human keratinocytes [25], but there are also contradictory reports stating that such induction does not occur [24,26]. IFN-γ (Th1 cytokine) has been reported to induce IL-33 production, which is mediated by activation of the epidermal growth factor receptor (EGFR), extracellular signal-regulated kinase (ERK), and p38 mitogen-activated protein kinase pathways [26,27]. Furthermore, since Janus kinase (JAK) 1/2 inhibitors suppress the IFN-γ-induced increase in IL-33 expression, the JAK1/2-signal transducer and activator of transcription (STAT) 1 pathway may be involved in this mechanism. IL-17 (Th17 cytokine) also reportedly induces IL-33 expression in keratinocytes via EGFR, ERK, p38 mitogen-activated protein kinase, and JAK/STAT1 pathways [28]. Stimulation with IL-4 and IL-13 (Th2 cytokines) has also been shown to induce IL-33 expression [29,30] (Figure 1). Since IL-4 and IL-13 induce activation of the JAK1/3–STAT6 pathway [31], it was expected that IL-4 may upregulate IL-33 expression via this pathway; however, analysis of keratinocytes stimulated with IL-4 revealed that treatment with JAK inhibitors did not affect IL-33 expression [30]. In addition, ERK inhibitors were reported to suppress the IL-4-induced expression of IL-33 [30]. Therefore, the mechanism behind IL-33 expression in keratinocytes may be more strongly dependent on ERK activation than on the JAK1/2–STAT6 pathway induced by IL-4 stimulation.

## 3. The Role of IL-33 in the Pathogenesis of Atopic Dermatitis (AD) and Psoriasis

### 3.1. Atopic Dermatitis (AD)

Atopic dermatitis (AD) affects 2–20% of the general population, and its rate varies by age and ethnicity [32]. It is a chronic inflammatory skin disease that causes eczematous lesions with intense pruritis [32]. The pathogenesis of AD is thought to occur via a complex interaction of type 2 immune responses, skin barrier dysfunction, and pruritus [33]. Skin barrier dysfunction is associated with decreased production of terminally differentiated molecules such as filaggrin (FLG) [33]. Abnormalities in the skin barrier also increase the colony formation of microorganisms such as *Staphylococcus aureus*, further exacerbating skin inflammation [34]. Severe pruritis decreases the patient’s quality of life and treatment satisfaction [35]. Scratching due to pruritis exacerbates skin inflammation by promoting cellular damage to the lesional skin [36].

Atopy is defined as the excessive production of immunoglobulin E (IgE) antibodies or a personal and/or family history of asthma, allergic rhinitis, allergic conjunctivitis, or AD [37]. Approximately 80% of AD patients show elevated serum IgE levels [38]. In contrast to patients with normal IgE levels and nonallergic intrinsic AD, mutations in the *FLG* gene and impaired skin barrier function are associated with disease severity in extrinsic AD patients with elevated IgE levels [39]. Recent genome-wide association studies have shown at least 19 significant susceptibility loci for AD, with T helper 2 (Th2) cytokines (KIF3A/IL-4/IL-13), IL-1 family receptors (IL1RL1/IL18R1/IL18RAP), and skin barrier proteins such as FLG highlighted as possible contributors [32].

Based on the pathological role of skin barrier function and immune abnormalities in AD, standard treatments include topical steroids, calcineurin inhibitors, JAK inhibitors, PDE4 inhibitors, and systemic therapies such as ultraviolet irradiation, cyclosporine, biologic agents, and oral JAK inhibitors [40]. These control the pathogenesis of AD by suppressing the signaling pathway of the type 2 immune response.

IL-33 has attracted attention as a cytokine that triggers the type 2 immune response. In a transgenic mouse model overexpressing IL-33 in the skin, dermatitis similar to AD was found to develop [41]. In mice lacking IL-33, AD-like lesions caused by the topical application of MC903 (calcipotriene: vitamin D analog) are not severe [42]. It was also previously observed that IL-33 is strongly expressed in the nuclei of keratinocytes in the lesions of AD patients [43]. The concentration of IL-33 in the serum of AD patients has also been found to be predominantly higher than that of normal subjects and psoriasis patients [44]. Keratinocytes damaged by scratching injury secrete IL-33 into the lesion, which stimulates ILC2 and induces the production of type 2 cytokines such as IL-5 and IL-13 [45]. In addition, Th2 cells stimulated by IL-33 produce IL-4, IL-13, and IL-31 [45]. IL-4, IL-13, IL-31, and IL-33 decrease filaggrin and loricrin expression in keratinocytes, resulting in skin barrier dysfunction [46,47,48]. Furthermore, IL-31 and IL-33 directly stimulate itch-sensing nerves, resulting in intense pruritis and scratch behavior [49].

IL-33 also contributes to the exacerbation of inflammation in allergic diseases by activating eosinophils, basophils, and mast cells; IL-33 potently induces eosinophilia, produces superoxide, upregulates the expression of adhesion molecules, and enhances cell survival [50]. IL-33 has also been reported to promote the migration, maturation, adhesion, and survival of basophils and mast cells, and induce the production of inflammatory cytokines such as IL-4 and IL-13 [51]. Taking these findings together, the inhibition of IL-33 production in keratinocytes may be of critical importance in the treatment of AD (Figure 2).

### 3.2. Psoriasis

Psoriasis is a chronic inflammatory skin disease that causes relatively well-defined erythematous plaques with scaling. Psoriatic lesions occur on the skin and joints, greatly compromising cosmetic appearance and quality of life [52]. The efficacy of treatment of psoriasis with TNF-α inhibitors [53], IL-17 inhibitors [53], and IL-23 inhibitors [54,55] as biological agents that suppress the function of specific cytokines has been demonstrated. However, psoriasis that is intractable to treatment with these agents has also been reported, suggesting that an immune axis other than TNF-α, IL-17, and IL-23 may be involved in the pathogenesis of psoriasis [56].

IL-33 expression has been found in the nuclei of keratinocytes in skin lesions of psoriasis patients [28]. Transcriptomic analysis has also shown that IL-33 mRNA and protein levels are increased in psoriatic lesions compared with the levels in nonlesions [57]. In addition, IL-33 levels in the serum of psoriasis patients are known to be higher than in healthy individuals [58,59], and IL-33 levels decrease when the skin lesions are relieved by the treatment of psoriasis [58]. In a mouse model of imiquimod-induced psoriasis, mice lacking epidermis-specific IL-33 developed milder psoriatic skin lesions than control mice [57]. In addition, the stimulation of human keratinocytes with IL-33 was found to produce the chemokines C-X-C motif chemokine ligand (CXCL) 1 and CXCL8, which recruit neutrophils, and CC chemokine ligand (CCL) 20, which recruits IL-17-producing T cells, to the epidermis [57]. Neutrophils and IL-17-producing T cells are important immune cells in psoriasis [36], so the above results suggest that IL-33 is also deeply involved in the pathogenesis of this condition. Activated dendritic cells (DCs) produce TNF-α and IL-23, which induce the differentiation and maturation of Th17 cells [53]. In addition, DCs matured by IL-33 induce differentiation into Th17 cells [60], which produce large amounts of IL-17A [52]; these in turn increase the proliferative capacity of keratinocytes and cause epidermal thickening in psoriasis [52]. In addition, IL-17A stimulates keratinocytes to produce the chemokine CCL20, which induces selective chemotaxis of DCs and Th17 cells via C-C chemokine receptor type 6 (CCR6) [52]. IL-17A also stimulates keratinocytes to produce CXCL8 and IL-36G, which induce neutrophil chemotaxis [36]. Thus, keratinocyte-derived IL-33 contributes to the pathogenesis of psoriasis by promoting the maturation and differentiation of Th17 cells by dendritic cells and the infiltration of neutrophils and IL-17-producing cells into the epidermis.

Furthermore, more than 60% of patients with moderate to severe psoriasis complain of pruritis [61], but the mechanism of pruritis in psoriasis remains unclear. Although IL-17A, IL-23, and TNF-α are key cytokines in psoriasis, their subcutaneous injection into mice did not induce scratching behavior [62,63]. Meanwhile, another study showed that IL-33 induced scratching behavior when injected subcutaneously [64]. Therefore, IL-33 is a cytokine that directly stimulates nerves to cause pruritis and may be related to pruritis in AD and psoriasis. However, unlike in AD, IL-31 is not associated with pruritis in psoriasis [65]. It has recently been proposed that there are two populations in the immunological background of psoriasis patients [66]. In these populations, another psoriasis population was identified that shared high levels of IL-17A and IL-17F, but also high levels of Th2 cytokines such as IL-4 and IL-13 [66]. This means that, in addition to the immune response by Th17, a Th2 immune response is also involved in the pathogenesis of psoriasis. Since IL-33 induces a Th2 immune-deviated response via ILC2 [45], IL-33 may be pathologically relevant in a population of Th17 plus Th2 cells involved in the immune response in psoriasis patients (Figure 2).

## 4. Structure and Signaling Pathway of IL-37

IL-37 (IL-1F7) is a member of the IL-1 family of cytokines. The gene encoding IL-37 is located on chromosome 2q12-13, very close to the regulatory region of the *IL-1α* and *IL-1β* genes [67]. This specific location may be important for the role of IL-37 in inhibiting inflammation. The size of the *IL-37* gene is 3617 bp, and its mRNA undergoes alternative splicing, resulting in five isoforms (a–e) [68]. Among these isoforms, IL-37b contains five exons and is the most complete of the isoforms, so most research has focused on it [68]. Meanwhile, IL-37c and IL-37e lack exon 4, which encodes a domain essential for the maintenance of IL-37 activity. Therefore, they are predicted to be nonfunctional [68]. The specific activities of IL-37a and IL-37d are still unknown, and there have been only limited studies on these specific IL-37 isoforms. In addition, no gene homologous to IL-37 has been identified in mice [69]. Therefore, experiments aimed at revealing roles of IL-37 in vivo require the generation of transgenic mice, such as those carrying the human *IL-37* gene.

IL-37 functions as an extracellular and intracellular cytokine. Exon 1 of IL-37 contains a site for cleavage by caspase-1. The precursor protein pro-IL-37 is cleaved by caspase-1 to form mature IL-37 [70]. When cytoplasmic pro-IL-37 is released extracellularly, it is converted to mature IL-37 by various proteases [70]. Extracellularly secreted mature IL-37 exerts its anti-inflammatory effects by acting on receptors associated with IL-18R. Specifically, IL-18 forms a complex composed of IL-18Rα and IL-18Rβ, which induces inflammation via the activation of MyD88. In this case, mature IL-37 binds to IL-18Rα and IL-1 receptor 8 (IL-1R8) to form an IL-18–IL-18Rα–IL-18Rβ complex that inhibits MyD88 activation and competitively suppresses signaling by IL-18 [70].

A portion of pro-IL-37 generated in the cytoplasm is cleaved by caspase-1 to become mature IL-37 [71]. Since IL-37 does not have a nuclear localization sequence, it requires other factors to translocate into the nucleus and regulate target gene expression [69]. After cleavage by caspase-1, IL-37 binds to Smad3 to form the IL-37–Smad3 complex [69]. The phosphorylation of Smad3 allows the nuclear translocation of IL-37 and suppresses inflammatory gene expression. The nuclear IL-37–Smad3 complex promotes the dephosphorylation of tyrosine phosphorylation-dependent signaling pathways such as MAPK, phosphoinositide 3-kinase (PI3K), NF-κB, and STAT3, leading to the inhibition of these signal transductions [72,73].

Thus, IL-37 exerts its anti-inflammatory effects both extracellularly and intracellularly. Extracellular IL-37 exerts its anti-inflammatory effects by competitively inhibiting the action of inflammatory IL-18. Meanwhile, intracellular IL-37 negatively regulates the expression of inflammation-related genes via Smad3.

## 5. The Role of IL-37 in the Pathogenesis of Atopic Dermatitis (AD) and Psoriasis

### 5.1. Atopic Dermatitis (AD)

Contact hypersensitivity (CHS) is one of the most important factors in the development of eczematous skin lesions [74]. Mice overexpressing human IL-37 show reduced auricular swelling and suppressed CHS after hapten application [74]. As a mechanism for this, it has been shown that IL-37 expression in dendritic cells suppresses CHS by inhibiting antigen-presenting ability and maturation and inducing regulatory T cells [74]. Furthermore, in an MC903-induced AD mouse model, mice expressing human IL-37b suppressed auricular swelling, pruritis, and the production of inflammatory cytokines and chemokines in the development of AD [4,75]. The induction of regulatory T cells has also been shown in this experimental model [75]. Subcutaneous injection of human IL-37b into the MC903-induced AD mouse model similarly attenuated auricular swelling and reduced the infiltration of basophils [75]. In an in vitro study, the production of basophil-derived IL-4 stimulated by thymic stromal lymphopoietin (TSLP), which is important in the pathogenesis of AD, was suppressed by human IL-37b [75]. This suggests that human IL-37b may improve the development of AD via basophils [75]. The significance of serum IL-37 production in AD patients is controversial, with one report showing that it correlated with the severity of the skin lesions [76] and another showing lower levels than in normal individuals [63]. Nevertheless, IL-37 expression in the lesions has been shown to be decreased compared with that in nonlesions [77,78].

Immunohistochemical analysis has shown that IL-37 is expressed in the granular layer of the epidermis and colocalizes with epidermal differentiation complex (EDC), including filaggrin [78] and loricrin [79], in healthy individuals. Also, analysis on the correlation between IL-37 expression and EDC in AD patients compared with healthy donors has revealed a positive correlation between the decrease in IL-37 expression with a decrease in filaggrin expression [78]. The mechanism behind this may involve IL-33, the major cytokine in AD. Since IL-33 and IL-33-induced IL-4 and IL-13 downregulate filaggrin and loricrin expression [46,47,48], it is assumed that IL-33 negatively regulate IL-37 expression via filaggrin and loricrin. In addition, as mentioned above, IL-33 produces chemokines such as CXCL1 and CXCL8, which induce neutrophil infiltration, and CCL20, which recruits Th17 cells, in human keratinocytes [57]. Since IL-17 can also contribute to reduced expression of filaggrin and loricrin [46], IL-33 may decrease IL-37 expression by inducing IL-17 production from neutrophils and Th17 cells.

These findings suggest that IL-37 suppresses Th2 immune responses in AD, and that the reduction of IL-37 is involved in the pathogenesis of AD (Figure 1).

### 5.2. Psoriasis

Human IL-37b inhibits the production of CXCL8, IL-6, and S100A7, which are important in the pathogenesis of psoriasis, in keratinocytes stimulated with inflammatory cytokines (TNF-α, IL-17A, IL-22, IL-1α, and Oncostatin-M) [80]. In addition, the injection of a plasmid containing human IL-37 improved the skin lesions with reduced IFN-γ expression in a mouse model of psoriasis in which epidermis-specific VEGF-A was overexpressed [80]. These results suggest that human IL-37 may be effective at reducing inflammation in psoriasis. However, in a mouse model with imiquimod-induced psoriasis, the injection of human IL-37b did not improve the dermatitis [81]. The concentration of IL-37 in the serum of psoriasis patients is low compared with that in healthy controls [81]. In addition, a positive correlation between serum IL-37 concentration and the severity of skin lesions has been reported [82]. Furthermore, tofacitinib, a JAK inhibitor, was found to improve the skin lesions in psoriasis patients with an increase of the epidermal expression of IL-37 [83]. Consistent with these results, transcriptomic analysis of psoriasis patients reportedly showed that IL-37 expression was decreased in skin lesions of psoriasis compared with that in healthy individuals [79].

Also, immunohistochemical analysis has shown that IL-37 expression in the granular layer of the epidermis is absent from psoriasis lesions [79]. Consistent with this, in vitro study has reported that IL-17A and IL-22 play roles in suppressing IL-37 expression [81]; however, it is still unclear whether IL-21 and IL-26, as well as IL-17 and IL-22 produced by Th17 cells, affect IL-37 expression. Given that IL-17A and IL-22 downregulate filaggrin and loricrin expression [46] and that IL-37 colocalizes with filaggrin and loricrin in keratinocytes [78,79], there is a possibility that IL-17A and IL-22 negatively regulate IL-37 expression via filaggrin and loricrin. Furthermore, TNF-α, a key cytokine in the pathogenesis of psoriasis, reportedly decreased filaggrin and loricrin expression in human keratinocytes [84]. Also, treatment of psoriasis patients with etanercept, an anti-TNF-α antibody, reversed filaggrin and loricrin downregulation [84]. These results indicate that treatment of psoriasis patients with biologics such as IL-17 inhibitors and TNF-α inhibitors may restore skin barrier dysfunction, preventing suppression of IL-37 expression in psoriasis.

IL-37 expression is modulated by inflammatory cytokines and TLR ligands [67]. Protein levels of IL-37 in peripheral blood mononuclear cells (PMBCs) and dendritic cells were increased by cytokines associated with the pathogenesis of psoriasis, such as IL-1β, IL-18, IFN-γ, and TNF-α [67]; however, in human keratinocytes, only TNF-α among these cytokines increased mRNA levels, but not the protein levels, of IL-37 [85]. Thus, the mechanism of IL-37 upregulation may differ between the cell types. It has been shown that human beta-defensin (hBD)-3, an antimicrobial peptide, increased mRNA and protein levels of IL-37 in human keratinocytes [85]. hBD-3-induced IL-37 upregulation was suppressed by treatment with neutralizing antibody of CCR6 and transfection of siRNA against CCR6, indicating that hBD-3 increases IL-37 expression via CCR6 [85]. Since hBDs, including hBD-3, are over-expressed in the epidermis of psoriasis patients [86], hBD-3 is likely to attenuate the development of psoriasis via IL-37 expression.

These findings suggest that IL-37 may serve as a therapeutic target for psoriasis (Figure 2).

## 6. Regulatory Mechanism of the IL-33–IL-37 Axis via Aryl Hydrocarbon Receptor (AHR)

AHR is a multidomain cytoplasmic protein belonging to the basic helix-loop-helix/per-Arnt-sim (bHLH/PAS) family of transcription factors [87]. AHR can bind exogenous (polycyclic aromatic hydrocarbons, dioxins, benzo[a]pyrene) or endogenous (6-formylindolo[3,2-b]carbazole: FICZ, kynurenine) ligands in the cytoplasm [87,88]. 2,3,7,8-Tetrachlorodibenzo-p-dioxin (TCDD) and benzo[a]pyrene are known to exhibit cellular toxic responses by activating AHR-mediated oxidative stress [89,90]. In fact, genes involved in the processing of toxic substances, pollutants, and endocrine disruptors were upregulated in the skin of AD patients, suggesting that inflammation is associated with the increased excretion of xenobiotics, pollutants, and endocrine disruptors in the skin of AD patients [91].

Meanwhile, AHR ligands that do not cause oxidative stress activate NRF2, which induces antioxidant enzymes and exerts anti-inflammatory effects [92,93]. Biological responses induced by AHR ligands vary depending on the ligand and cell type, and this diversity makes the AHR signaling complex [94].

AHR in an inactive state is located in the cytoplasm and forms complexes with molecular chaperones such as heat shock protein 90 (HSP90) and co-chaperones such as p23 and AHR-interacting protein (AIP) [95]. In the presence of ligands, AHR translocates into the nucleus and interacts with AHR nuclear transporter (ARNT) and AHR repressor (AHRR) via the PAS domain [95]. AHR forms a heterodimeric complex with ARNT upon translocation into the nucleus and binds to drug response elements (DREs) in the promoter region of its target genes [95]. It has been reported that the transcriptional activity of DRE sequences is regulated by the methylation patterns around such sequences, indicating that AHR may induce adaptive cellular responses to AHR ligands by modulating the methylation around DRE sequences [96].

Two DREs have been identified in the promoter region of human IL-33 (−726 bp upstream of IL-33 exon 1c: DRE1 and −470 bp: DRE2), and electrophoretic mobility shift assay (EMSA) and chromatin immunoprecipitation (ChIP) assay have confirmed AHR recruitment to DRE1 and DRE2, indicating that IL-33 is transcriptionally regulated by AHR [97]. In addition, luciferase assay has revealed that mutations in DRE1 increase the activity of luciferase, while mutations in DRE2 decrease it [97]. This suggests that DRE1 suppresses IL-33 expression and DRE2 increases it in an AHR ligand-dependent manner [97]. Since the recruitment of AHR to DREs depends on the ligand that binds to AHR, DRE1 and DRE2 may be involved in the mechanism of the biological response in which IL-33 expression depends on the AHR ligand; however, further studies are needed to clarify the mechanism by which DRE1 and DRE2 regulate IL-33 promoter activity.

NF-κb response elements that are activated via the Toll-like receptor (TLR)4 signaling pathway were also found in the promoter of IL-33 [98]. Stimulation of THP-1 macrophages with particulate matter (PM)2.5, an environmental contaminant and a ligand of AHR, was reported to increase IL-33 production [98]. This PM2.5-induced IL-33 upregulation was also shown to be attenuated by AHR knockdown or TLR4 mutation [98]. This indicates that PM2.5-induced IL-33 upregulation is dependent on the activation of AHR and TLR4. However, whereas TLR4 mutation almost completely attenuated PM2.5-induced IL-33, the inhibitory effect of AHR knockout (KO) on IL-33 upregulation induced by PM2.5 was partial [98]. Thus, PM2.5 may be both an AHR ligand and a TLR4 ligand. The AHR agonist TCDD also induced IL-33 production in THP-1 macrophages [97]. TCDD-induced IL-33 upregulation was canceled by pretreatment with the AHR antagonist CH223191, indicating that TCDD induces IL-33 production in an AHR-dependent manner [97]. Given that PM2.5 and TCDD are environmental pollutants, air pollution may increase IL-33 production in macrophages, contributing to the exacerbation of inflammation. In fact, the expression of IL-33 induced by AHR ligands has been found to vary among cell types. In MCF7 cells, a human breast cancer line, and U937 cells, a human lymphoma cell line, IL-33 expression increased in the presence of AHR ligands [97], but this did not occur in HepG2 cells, a human hepatoma cell line, or A549 cells, a human lung cancer cell line [97]. In THP-1 macrophages, the AHR ligand FICZ was shown to increase IL-33 expression [97], but FICZ suppressed IL-33 expression in human keratinocytes [99].

It has also been reported that the production of IL-33 is suppressed by agents that act on AHR, such as the soybean extract tar glyteer [30], propionate [99], and tapinarof [30,100], in human keratinocytes. Tapinarof, a drug categorized as a therapeutic AHR-modulating agent (TAMA), has been developed as a topical treatment for both AD and psoriasis [101,102]. Tapinarof has been shown to upregulate filaggrin and loricrin in human keratinocytes [103]. Tapinarof cream was approved for use in psoriasis by the Food and Drug Administration (FDA) in May 2022 after its efficacy was confirmed in a clinical trial [104]. Also, a Phase II trial in adults with AD showed that tapinarof displayed a significant improvement in skin scores and pruritis after 12 weeks of topical application [101]. A Phase III trial for AD is currently underway in Europe, the US, and Japan.

The knockdown of AHR in human keratinocytes increased IL-33 expression by itself [30]. In addition, the knockdown of AHR canceled the inhibitory effects of glyteer and propionate on IL-33 expression [30,99]. Moreover, in a mouse model of MC903-induced AD, topical application of propionate suppressed the increase in IL-33 expression, which was not observed in AHR KO mice [99]. These results indicate that AHR exerts a negative regulatory effect on *IL-33* gene transcription in keratinocytes. The mechanism by which AHR ligands exert different effects is not well understood but may be due to ligand-specific changes in the AHR complex [94].

In addition to AHR, IL-33 expression is also regulated by ovo-like 1 (OVOL1) [30], a transcription factor downstream of AHR [105]. OVOL1 is one of the disease susceptibility genes in AD and regulates the expression of proteins involved in skin barrier function [105,106,107]. The knockdown of OVOL1 in human keratinocytes alone increases IL-33 expression [30]. Furthermore, the inhibitory effect of glyteer on IL-33 was abolished not only by AHR but also by the knockdown of OVOL1 [30]. Conditional KO mice with specific deletion of Ovol1 in the epidermis showed IL-33 upregulation in the skin [108]. Although the mechanism by which OVOL1 regulates IL-33 expression in human keratinocytes is still unclear, it has been shown that the knockdown of OVOL1 enhances the phosphorylation of ERK1/2, one of the MAPKs [30]. Since the phosphorylation of ERK1/2 is involved in IL-33 upregulation [26,27,28,29], it is expected that increased activity of the ERK1/2 signaling pathway induced by the downregulation of OVOL1 would increase IL-33 expression in human keratinocytes [30].

Many reports have been published on the compounds from Chinese herbal medicine and the flavonoids that mitigate allergic symptoms and suppress IL-33 production [109], but their molecular mechanisms remain largely unknown. In Chinese herbal medicine, Sho-sai-ko-to was reported to relieve OVA-induced asthma symptoms and inhibit IL-33 production in bronchoalveolar lavage fluid in a murine asthma model [110]. Sho-sai-ko-to contains Bupleuri Radix, Scutellariae Radix, and Ginseng Radix, which have been shown to act on AHR [111,112,113]. Among the flavonoids, luteolin and apigenin have been reported to inhibit IL-33 production [114], and these flavonoids have also been reported to act on AHR [115,116]. Therefore, the inhibitory effects of Chinese herbal medicine and flavonoids on IL-33 production may involve the AHR signaling pathway.

As we have shown, IL-33 and IL-37 are highly expressed in the epidermis of AD and psoriasis. Therefore, the expression of IL-33 might be closely related to that of IL-37. It has also been shown that IL-33 is negatively regulated by IL-37 in human keratinocytes [100]. A microarray analysis of normal human keratinocytes with siRNA-induced IL-37 knockdown showed that this knockdown increased IL-33 mRNA and protein levels [100]. Furthermore, the knockdown of IL-37 induced the phosphorylation of MAPK, suggesting that the activation of MAPK by IL-37 knockdown may have contributed to the increased expression of IL-33 [100]. As mentioned above, IL-37 colocalizes with filaggrin [78] and loricrin [79]. The activation of AHR is a potent inducer of filaggrin and loricrin, which in turn promotes keratinocyte differentiation [31,103]. Thus, agents acting on AHR may reinforce IL-37 expression by increasing filaggrin and loricrin. Furthermore, the expression of IL-37 in keratinocytes is transcriptionally regulated by AHR [100]. We stimulated keratinocytes with tapinarof and *Galactomyces* ferment filtrate (GFF) acting on AHR and analyzed IL-37 expression. The results showed that tapinarof and GFF increased the mRNA and protein levels of IL-37. Moreover, siRNA-induced knockdown of AHR reversed the elevation of IL-37 mRNA and protein levels induced by tapinarof and GFF, indicating that AHR exerts a direct regulatory effect on IL-37 expression in keratinocytes [100] (Figure 3).

To date, only a few reports of compounds that induce IL-37 expression have been published. PG102, a plant-derived substance extracted from *Actinidia arguta*, has been reported to increase IL-37 expression in human keratinocytes [117]. *A. arguta* extract contains many polyphenols, among which quercetin and kaempferol derivatives have been identified as major components [118,119]. Since quercetin and kaempferol have been shown to activate AHR [120], it is possible that PG102 may also increase IL-37 expression via AHR. Ultraviolet (UV) irradiation also increases IL-37 expression in the skin [121]. It has also been reported that UV irradiation of the skin causes intracellular tryptophan to undergo a conformational change in the endogenous AHR ligand FICZ [122]. Thus, the increase in IL-37 expression upon UV irradiation may be mediated by an AHR-dependent mechanism.

## 7. Conclusions

IL-33, which triggers inflammation, and IL-37, which suppresses it, are coordinately regulated. IL-33 expression is partially dependent on MAPK activation, and IL-37 is responsible for suppressing MAPK. In addition, the Th2 immune response via IL-33 downregulates skin barrier function proteins such as filaggrin and loricrin, which is co-expressed with IL-37. In AD and psoriasis, there is an imbalance of the IL-33–IL-37 axis, and pathogenesis is likely to occur via increased IL-33 and decreased IL-37. The role of IL-33/IL-37 interactions in inflammatory skin diseases has been reported [5]. However, the factors that regulate the balance between IL-33 and IL-37 expression are still unclear. As such, AHR-mediated regulation of the IL-33–IL-37 axis may lead to new therapeutic strategies for AD and psoriasis (Figure 4).

## Figures and Tables

**Figure 1 ijms-24-14633-f001:**
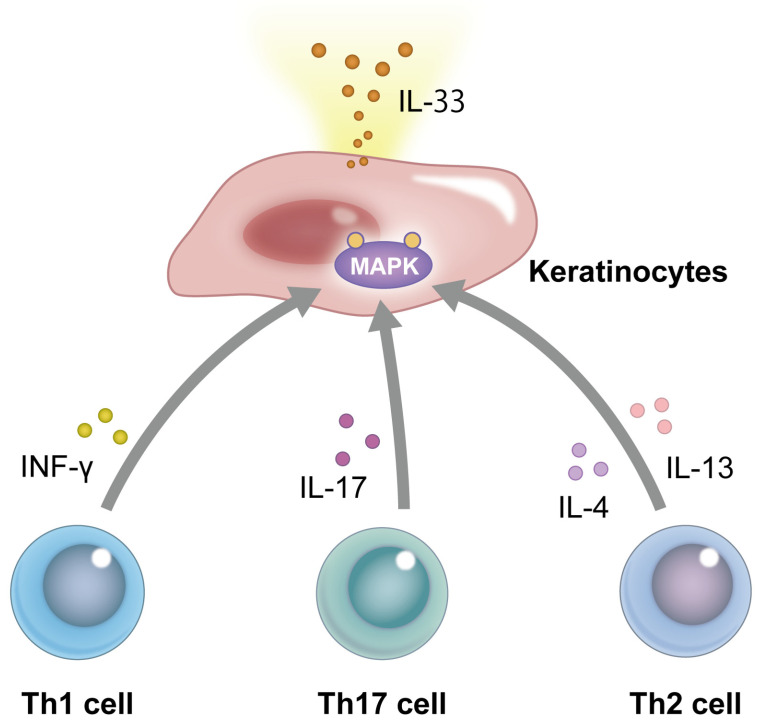
Production of IL-33 in human keratinocytes. IFN-γ (Th1), IL-17 (Th17), IL-4, and IL-13 (Th2) induce the production of IL-33 via activation of mitogen-activated protein kinase (MAPK) in human keratinocytes.

**Figure 2 ijms-24-14633-f002:**
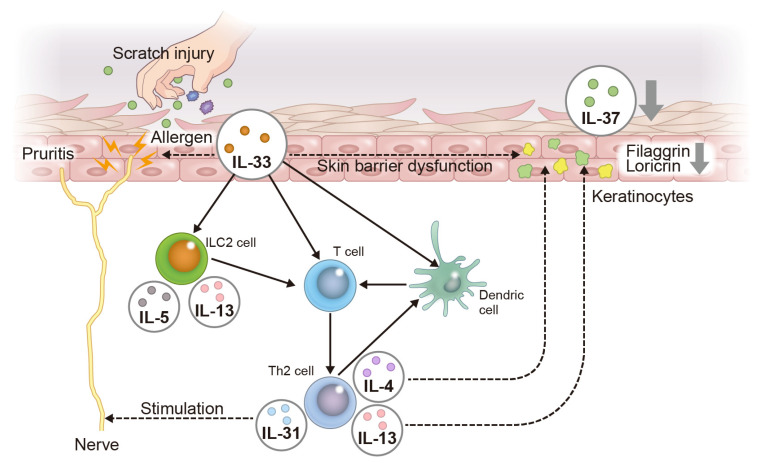
Pathogenesis of AD focusing on the IL-33–IL-37 axis in keratinocytes. Scratch injury due to pruritis and allergens produces IL-33 in keratinocytes. IL-33 directly stimulates itch-sensing nerves to induce pruritis. IL-33 activates ILC2, leading to the upregulation of IL-5 and IL-13, which involves a Th2-deviated immune response. Th2 cells amplify the Th2 immune response through interaction with dendritic cells. Th2 cells produce IL-4 and IL-13, resulting in skin barrier dysfunction, such as reduced filaggrin and loricrin expression. IL-31 derived from Th2 cells stimulates itch-sensing nerves, contributing to pruritis. The downregulation of filaggrin and loricrin expression induced by IL-33, IL-4, and IL-13 negatively regulates IL-37 expression in keratinocytes.

**Figure 3 ijms-24-14633-f003:**
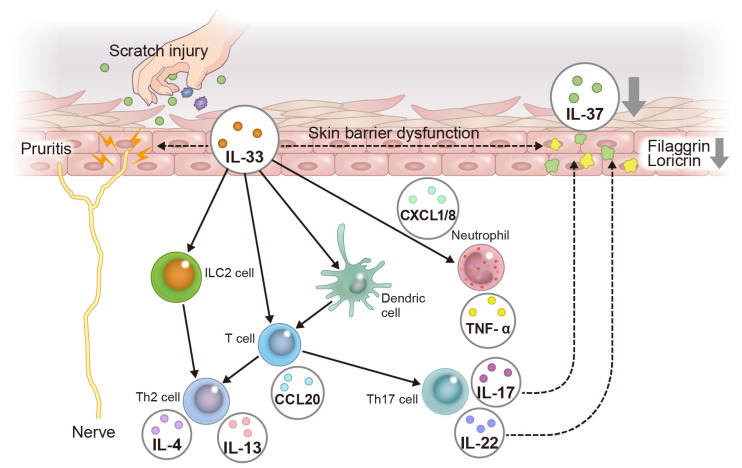
Pathogenesis of psoriasis focusing on the IL-33–IL-37 axis in keratinocytes. Psoriasis patients with pruritis produce IL-33 in keratinocytes by scratch injury. IL-33 directly stimulates itch-sensing nerves to induce pruritis. IL-33 tends to induce a Th2 immune response in psoriasis as well as AD. IL-33 also induces the production of CCL20, which recruits Th17 cells, and CXCL8, which recruits neutrophils, to the epidermis. The accumulation of Th17 cells and neutrophils contributes to the development of psoriasis. IL-33, IL-17A, and IL-22 downregulate filaggrin and loricrin expression, which contributes to reduced IL-37 expression in keratinocytes.

**Figure 4 ijms-24-14633-f004:**
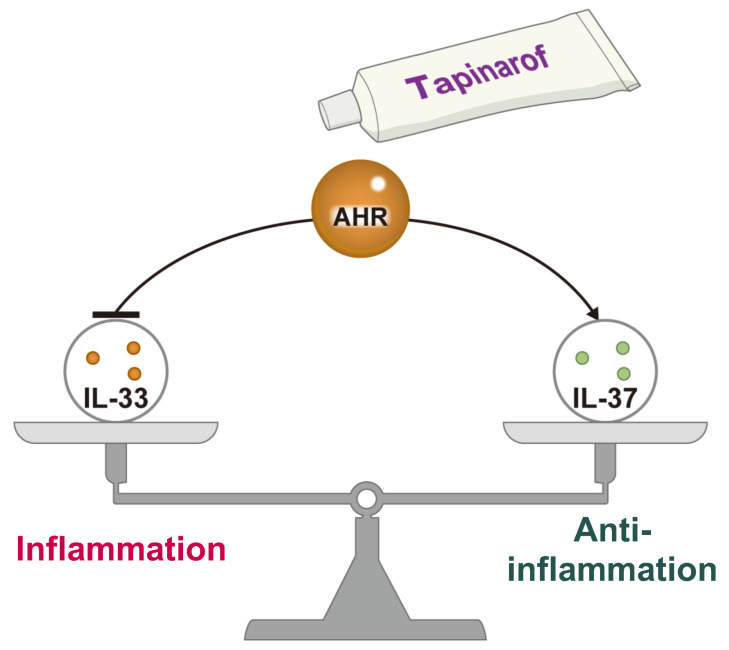
Balance between inflammation and anti-inflammation via the IL-33–IL-37 axis in AD and psoriasis. IL-33, which triggers inflammation, and IL-37, which suppresses it, are coordinately regulated. AHR modulation by tapinarof inhibits IL-33 and induces IL-37, preventing the development of AD and psoriasis.

## Data Availability

The data that support the findings of this study are available from the corresponding author, [G.T.], upon reasonable request.

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
