# Peer review of "Regulatory Mechanism of the IL-33–IL-37 Axis via Aryl Hydrocarbon Receptor in Atopic Dermatitis and Psoriasis"

_ijms, 2023, doi:10.3390/ijms241914633_

Round 1
Reviewer 1 Report
Dear authors,
I read your manuscript with great interest. It is very well-written and easy to follow. Very comprehensive and interesting review on the role of IL-33, IL-37 axis in the pathophysiology of atopic dermatitis and psoriasis as well as the role of aryl hydrocarbon receptor.
I find the manuscipt in its current form suitable for publication.
Author Response
Thank you very much for the comment.
Reviewer 2 Report
I have reviewed the paper entitled: “Regulatory mechanism of the IL-33–IL-37 axis via aryl hydrocarbon receptor in atopic dermatitis and psoriasis”. In this review, the authors mention some mechanisms related to the IL-33/IL-37 axis and its participation in the pathogenesis of skin diseases such as atopic dermatitis and psoriasis. I comment follow:
1. In section 2, a figure could be included representing the activation mechanisms of IL-33 by the different subtypes of Th cells 1/2/17.
2. In section 3. Increasing the description of AD, I consider that very little is explained about the disease.
3. In paragraphs 88-103, improve the wording since some paragraphs sound very repetitive. In addition, I consider that it failed to mention what happens with other cells that express the ST2 receptor, that have been involved in allergic processes, and that are known to respond to stimulation by IL-33 as basophils, eosinophils and mast cells in AD.
4. From paragraphs 108-135, more should be said about the mechanisms of Th17 cells induced by IL-33, what is its importance, specific mechanisms of pathogenesis in psoriasis, the cytokines produced by Th17, what other cells they activate that may participate in psoriasis, what about the innate immune response induced by IL-33 or IL-17.
5. In section 5.1, it is mentioned that IL-33 downregulates filaggrin and loricrin and that this downregulation in turn inhibits IL-37 production. Is this the only possible mechanism described for IL-37 inhibition in AD or are there others?
6. In section 5.2, they mention that IL-17A and IL-22 inhibit the production of IL-37, Are there reports that mention other cytokines produced by Th17 cells such as IL-21 that also suppress the production of IL-37?
7. What different information does your review provide with respect to a review recently published in this journal by Borgia, F.; Custurone, P.; Li Pomi, F.; Vaccaro, M.; Alessandrello, C.; Gangemi, S. IL-33 and IL-37: A Possible Axis in Skin and Allergic Diseases. Int. J. Mol. Sci. 2023, 24, 372. https://doi.org/10.3390/ijms24010372
Author Response
- In section 2, a figure could be included representing the activation mechanisms of IL-33 by the different subtypes of Th cells 1/2/17.
Thank you very much for the comment. We have made a new figure showing IL-33 production in keratinocytes stimulated by Th1, 2, and 17 cells.
- In section 3. Increasing the description of AD, I consider that very little is explained about the disease.
Thank you very much for the comment. We have added a description of the disease of atopic dermatitis as follows:
(Lines 89–125)
Atopic dermatitis (AD) affects 2%–20% of the general population and its rate varies by age and ethnicity [32]. It is a chronic inflammatory skin disease that causes eczematous lesions with intense pruritis [32]. The pathogenesis of AD is thought to occur via a complex interaction of type 2 immune responses, skin barrier dysfunction, and pruritus [33]. Skin barrier dysfunction is associated with decreased production of terminally differentiated molecules such as filaggrin (FLG) [33]. Abnormalities in the skin barrier also increase colony formation of microorganisms such as Staphylococcus aureus, further exacerbating skin inflammation [34]. Severe pruritis decreases the patient’s quality of life and treatment satisfaction [35]. Scratching due to pruritis exacerbates skin inflammation by promoting cellular damage to the lesional skin [36].
Atopy is defined as the excessive production of immunoglobulin E (IgE) antibodies or a personal and/or family history of asthma, allergic rhinitis, allergic conjunctivitis, or AD [37]. Approximately 80% of AD patients show elevated serum IgE levels [38]. In contrast to patients with normal IgE levels and non-allergic intrinsic AD, mutations in the FLG gene and impaired skin barrier function are associated with disease severity in extrinsic AD patients with elevated IgE levels [39]. Recent genome-wide association studies have shown at least 19 significant susceptibility loci for AD, with T helper 2 (Th2) cytokines (KIF3A/IL-4/IL-13), IL-1 family receptors (IL1RL1/IL18R1/IL18RAP), and skin barrier proteins such as FLG highlighted as possible contributors [32].
Based on the pathological role of skin barrier function and immune abnormalities in AD, standard treatments include topical steroids, calcineurin inhibitors, JAK inhibitors, and PDE4 inhibitors, and systemic therapies such as ultraviolet irradiation, cyclosporine, biologic agents, and oral JAK inhibitors [40]. These control the pathogenesis of AD by suppressing the signaling pathway of the type 2 immune response.
- In paragraphs 88-103, improve the wording since some paragraphs sound very repetitive. In addition, I consider that it failed to mention what happens with other cells that express the ST2 receptor, that have been involved in allergic processes, and that are known to respond to stimulation by IL-33 as basophils, eosinophils, and mast cells in AD.
Thank you very much for the comment. We have rewritten the sections noted. We have also described the effects of IL-33 on eosinophils, basophils, and mast cells as follows:
(Lines 113–131)
IL-33 has attracted attention as a cytokine that triggers the type 2 immune response. In a transgenic mouse model overexpressing IL-33 in the skin, dermatitis similar to AD was found to develop [41]. In mice lacking IL-33, AD-like lesions caused by the topical application of MC903 (calcipotriene: vitamin D analog) are not severe [42]. It was also previously observed that IL-33 is strongly expressed in the nuclei of keratinocytes in lesions of AD patients [43]. The concentration of IL-33 in the serum of AD patients has also been found to be predominantly higher than that of normal subjects and psoriasis patients [44]. Keratinocytes damaged by scratching injury secrete IL-33 into the lesion, which stimulates ILC2 and induces the production of type 2 cytokines such as IL-5 and IL-13 [45]. In addition, Th2 cells stimulated by IL-33 produce IL-4, IL-13, and IL-31 [45]. IL-4, IL-13, IL-31, and IL-33 decrease filaggrin and loricrin expression in keratinocytes, resulting in skin barrier dysfunction [46–48]. Furthermore, IL-31 and IL-33 directly stimulate itch-sensing nerves, resulting in intense pruritis and scratch behavior [49].
IL-33 also contributes to the exacerbation of inflammation in allergic diseases by activating eosinophils, basophils, and mast cells; IL-33 potently induces eosinophilia, produces superoxide, upregulates the expression of adhesion molecules, and enhances cell survival [50]. IL-33 has also been reported to promote migration, maturation, adhesion, and survival of basophils and mast cells, and induce the production of inflammatory cytokines such as IL-4 and IL-13 [51].
- From paragraphs 108-135, more should be said about the mechanisms of Th17 cells induced by IL-33, what is its importance, specific mechanisms of pathogenesis in psoriasis, the cytokines produced by Th17, what other cells they activate that may participate in psoriasis, what about the innate immune response induced by IL-33 or IL-17.
Thank you for the comment. We have described the effects of IL-33 on dendritic cells and detailed the mechanisms by which IL-33 shapes the pathogenesis of psoriasis via activation of dendritic cells, neutrophils, and IL-17-producing cells. Based on this, we have modified the section as follows:
(Lines 155–166)
Activated dendritic cells (DCs) produce TNF-α and IL-23, which induce the differentiation and maturation of Th17 cells [53]. In addition, DCs matured by IL-33 induce differentiation into Th17 cells [61], which produce large amounts of IL-17A [52]; these in turn increase the proliferative capacity of keratinocytes and cause epidermal thickening in psoriasis [52]. In addition, IL-17A stimulates keratinocytes to produce chemokine CCL20, which induces selective chemotaxis of DCs and Th17 cells via C-C chemokine receptor type 6 (CCR6) [52]. IL-17A also stimulates keratinocytes to produce CXCL8 and IL-36G, which induce neutrophil chemotaxis [36]. Thus, keratinocyte-derived IL-33 contributes to the pathogenesis of psoriasis by promoting the maturation and differentiation of Th17 cells by dendritic cells and the infiltration of neutrophils and IL-17-producing cells into the epidermis.
- In section 5.1, it is mentioned that IL-33 downregulates filaggrin and loricrin and that this downregulation in turn inhibits IL-37 production. Is this the only possible mechanism described for IL-37 inhibition in AD or are there others?
Thank you very much for the comment. As you have pointed out, we believe that skin barrier dysfunction caused by IL-33 is an important mechanism involved in the reduction of IL-37. Furthermore, IL-33 produces chemokines such as CXCL1 and CXCL8, which induce neutrophil infiltration, and CCL20, which recruits Th17 cells, in human keratinocytes. Thus, IL-33 production may decrease IL-37 expression by inducing IL-17 production derived from neutrophils and Th17 cells. Based on this, we have amended part of section 5.1 as follows:
(Lines 249–253)
In addition, as mentioned above, IL-33 produces chemokines such as CXCL1 and CXCL8, which induce neutrophil infiltration, and CCL20, which recruits Th17 cells, in human keratinocytes [58]. Since IL-17 can also contribute to reduced expression of filaggrin and loricrin [46], IL-33 may decrease IL-37 expression by inducing IL-17 production from neutrophils and Th17 cells.
- In section 5.2, they mention that IL-17A and IL-22 inhibit the production of IL-37, Are there reports that mention other cytokines produced by Th17 cells such as IL-21 that also suppress the production of IL-37?
Thank you very much for the comment. The regulatory mechanism of IL-37 by cytokines is less clear; we searched the literature for whether IL-21 and IL-26 produced by Th17 cells, in addition to IL-17 and IL-22, affect IL-37 expression, but this is still unclear. Based on this, we amended part of section 5.2 as follows:
(Lines 274–276)
Consistent with this, in vitro study has reported that IL-17A and IL-22 play roles in suppressing IL-37 expression [83]; however, it is still unclear whether IL-21 and IL-26, as well as IL-17 and IL-22 produced by Th17 cells, affect IL-37 expression.
- What different information does your review provide with respect to a review recently published in this journal by Borgia, F.; Custurone, P.; Li Pomi, F.; Vaccaro, M.; Alessandrello, C.; Gangemi, S. IL-33 and IL-37: A Possible Axis in Skin and Allergic Diseases. Int. J. Mol. Sci. 2023, 24, 372. https://doi.org/10.3390/ijms24010372
Thank you very much for the comment. The introduced review, as well as this review, describes the role of IL-33/IL-37 interactions in inflammatory skin diseases. This review contains more advanced findings than the introduced review because it describes the regulatory mechanisms of IL-33 and IL-37 in more detail from an AHR perspective and refers to therapeutic strategies targeting the AHR. The conclusions section has been added considering this point as follows:
(Lines 439–441)
The role of IL-33/IL-37 interactions in inflammatory skin diseases has been reported [5]. However, the factors that regulate the balance between IL-33 and IL-37 expression are still unclear.
Reviewer 3 Report
This is a highly complex topic requiring more input from authors to make it more comprehensive. The interaction of AHR and cytokines is a very relevant topic for skin disease and other chronic diseases.
1. Maybe authors can stress a bit more the importance of the source tissue and cellular composition. The skin has different cells with distinct expression profiles (PMID: 25545474) similarly the blood transcriptome can detect even meningitis profiles (PMID: 23515576). This is something authors should discuss more when reviewing the interaction of different transcripts. What is the source of the samples and where s the pathology?
2. AD and psoriasis have similarities and differences in molecular patterns. The authors clearly discuss some intracellular pathways. I recommend considering also other pathways that are relevant for skin pathologies (PMID: 18514490) and papers where the direct comparison of AD has been described (PMID: 28899689)
3. Findings with AHR are interesting, but the authors do not discuss the studies where the methylation of AHR has been different but without any effect on the expression of AHR like in the paper PMID: 26348578
I feel that the story is compelling and important to address, but it requires more comprehensive approach
Author Response
- Maybe authors can stress a bit more the importance of the source tissue and cellular composition. The skin has different cells with distinct expression profiles (PMID: 25545474) similarly the blood transcriptome can detect even meningitis profiles (PMID: 23515576). This is something authors should discuss more when reviewing the interaction of different transcripts. What is the source of the samples and where is the pathology?
The skin has a variety of component cells, and as you point out, transcriptomic analysis has shown that there are genes and pathways that are characteristic of or specific to melanocytes compared with epidermal cells and fibroblasts. In this review, we summarize the relationship between IL-33 and IL-37 in terms of histopathological findings and cytokine network, noting that keratinocytes are the major source of IL-33 and IL-37 and that they play an important role in the pathogenesis of atopic dermatitis and psoriasis. Based on this, we added a sentence in section 6 on the Regulatory mechanism of the IL-33–IL-37 axis via aryl hydrocarbon receptor (AHR) as follows:
(Lines 403–404)
As we have shown, IL-33 and IL-37 are highly expressed in the epidermis of AD and psoriasis. Therefore, the expression of IL-33 might be closely related to that of IL-37.
- AD and psoriasis have similarities and differences in molecular patterns. The authors clearly discuss some intracellular pathways. I recommend considering also other pathways that are relevant for skin pathologies (PMID: 18514490) and papers where the direct comparison of AD has been described (PMID: 28899689)
Thank you very much for the comment. We reviewed the article PMID: 18514490 to examine the signaling pathways associated with vitiligo vulgaris, atopic dermatitis, and psoriasis. However, MITF, CREB1, p38, USF1, PIK3CB (PI3K), RPS6KB1, LEF1, and BCL2 were not directly relevant molecules for this review.
It is very interesting that genes related to xenobiotic metabolism are upregulated in the skin of patients with atopic dermatitis (PMID: 28899689); given that the AHR plays a central role in xenobiotic metabolism, we believe that this will further clarify the role of the AHR in the pathogenesis of atopic dermatitis. We cited the paper (PMID: 28899689) and amended section 6 on the Regulatory mechanism of the IL-33–IL-37 axis via aryl hydrocarbon receptor (AHR) as follows:
(Lines 309–313)
In fact, genes involved in the processing of toxic substances, pollutants, and endocrine disruptors were upregulated in the skin of AD patients, suggesting that inflammation is associated with increased excretion of xenobiotics, pollutants, and endocrine disruptors in the skin of AD patients [93].
- Findings with AHR are interesting, but the authors do not discuss the studies where the methylation of AHR has been different but without any effect on the expression of AHR like in the paper PMID: 26348578
Thank you very much for the comment. We have added that, in addition to its role as a transcription factor, AhR regulates signaling pathways through an epigenetic mechanism in which it functions as a methylation reader, methylation modulator, and exhibits multifunctional properties as follows:
(Lines 324–327)
It has been reported that the transcriptional activity of DRE sequences is regulated by methylation patterns around DRE sequences, indicating that AHR may induce adaptive cellular responses to AHR ligands by modulating methylation around DRE sequences [98].
Reviewer 4 Report
The article is very interesting and well written, and the topic chosen by the authors is important to the literature.
I would have some major and minor revisions for the authors
1) In the introduction part the authors have chosen outdated references, I leave two recent reviews in the literature where they talk about the role of AHR not only in psoriasis or atopic dermatitis but also in other chronic inflammatory diseases with HS or acne, I ask the authors to elaborate on this in the introduction part
- Napolitano M, Fabbrocini G, Martora F, Picone V, Morelli P, Patruno C. Role of Aryl Hydrocarbon Receptor Activation in Inflammatory Chronic Skin Diseases. Cells. 2021;10(12):3559. Published 2021 Dec 16. doi:10.3390/cells10123559.
- Cardinali G, Flori E, Mastrofrancesco A, et al. Anti-Inflammatory and Pro-Differentiating Properties of the Aryl Hydrocarbon Receptor Ligands NPD-0614-13 and NPD-0614-24: Potential Therapeutic Benefits in Psoriasis. Int J Mol Sci. 2021;22(14):7501. Published 2021 Jul 13. doi:10.3390/ijms22147501.
2) In section 3.1 and 3.2, the authors could also add a few sentences about the treatment available for these two diseases
I leave some interesting articles for the authors to use
- DOI:10.2147/CCID.S364640
- DOI: 10.1080/14712598.2022.2132143
- DOI: 10.1159/000527007
3) There are typos in the text, I recommend revising it entirely especially the numbering of references should be checked
4) The paragraph AD would be preferable not to use the abbreviation
Minor editing of English language required
Author Response
- In the introduction part the authors have chosen outdated references, I leave two recent reviews in the literature where they talk about the role of AHR not only in psoriasis or atopic dermatitis but also in other chronic inflammatory diseases with HS or acne, I ask the authors to elaborate on this in the introduction part
Thank you very much for the comment. We cited your two recommended articles and amended part of the introduction as follows:
(Lines 34–41)
It has also been shown that the aryl hydrocarbon receptor (AHR), a receptor for dioxins, is involved in the pathogenesis of inflammatory skin diseases such as atopic dermatitis (AD), psoriasis [7,8], hidradenitis suppurativa (HS), and acne [9]. AHR contributes to the pathogenesis of these inflammatory skin diseases by regulating the production of inflammatory cytokines such as TNF-α and IL-8 [9] and the Th17 cytokine network [9]. In addition, it has been reported that the ligands of AHR have two opposing sets of properties, one that aggravates inflammation and one that suppresses inflammation [10].
- In section 3.1 and 3.2, the authors could also add a few sentences about the treatment available for these two diseases
Thank you very much for the comment. We cited the articles you recommended and amended the part as follows:
(Lines 108–112)
Based on the pathological role of skin barrier function and immune abnormalities in AD, standard treatments include topical steroids, calcineurin inhibitors, JAK inhibitors, and PDE4 inhibitors, and systemic therapies such as ultraviolet irradiation, cyclosporine, biologic agents, and oral JAK inhibitors [40]. These control the pathogenesis of AD by suppressing the signaling pathway of the type 2 immune response.
(Lines 137–142)
The efficacy of treatment of psoriasis with TNF-α inhibitors [53], IL-17 inhibitors [53], and IL-23 inhibitors [54,55] as biological agents that suppress the function of specific cytokines has been demonstrated. However, psoriasis that is intractable to treatment with these agents has also been reported, suggesting that an immune axis other than TNF-α, IL-17, and IL-23 may be involved in the pathogenesis of psoriasis [56].
- There are typos in the text, I recommend revising it entirely especially the numbering of references should be checked.
Thank you very much for the comment. We have corrected the typos and checked the numbering of references.
4) The paragraph AD would be preferable not to use the abbreviation
Thank you very much for the comment. We have changed “AD” “to atopic dermatitis.”
Round 2
Reviewer 2 Report
The authors have made the modifications suggested by the reviewers. The manuscript reads much better. I consider that the article should be accepted for publication
Reviewer 4 Report
Article is improved
In my opinion is suitable for publication